# Third-Person Visual Imitation Learning via Decoupled Hierarchical Controller

**Pratyusha Sharma**
MIT

**Deepak Pathak**
Facebook AI Research

**Abhinav Gupta**
CMU

## Abstract

We study a generalized setup for learning from demonstration to build an agent that can manipulate novel objects in unseen scenarios by looking at only a single video of human demonstration from a third-person perspective. To accomplish this goal, our agent should not only learn to understand the intent of the demonstrated third-person video in its context but also perform the intended task in its environment configuration. Our central insight is to enforce this structure explicitly during learning by decoupling what to achieve (intended task) from how to perform it (controller). We propose a hierarchical setup where a high-level module learns to generate a series of first-person sub-goals conditioned on the third-person video demonstration, and a low-level controller predicts the actions to achieve those sub-goals. Our agent acts from raw image observations without any access to the full state information. We show results on a real robotic platform using Baxter for the manipulation tasks of pouring and placing objects in a box. Project video is available at https://pathak22.github.io/hierarchical-imitation/.

## 1 Introduction

Humans have an extraordinary ability to perform complex operations by *watching others*. How do we achieve this? Imitation requires inferring the goal/intention of the other person one is trying to imitate, translating these goals into one's own context, mapping the third-person's actions to first-person actions, and then finally using these translated goals and mapped actions to perform low-level control. For example, as shown in Figure 1, imitating the pouring task not only involves understanding how to change object states (tilt glass on top of another glass), but also imagining how to adapt goals to novel objects in scene followed by low-level control to accomplish the task.

As one can imagine, simultaneously learning these functions is extremely difficult. Therefore, most of the classical work in robotics has focused on a much-restricted version of the problem. One of the most common setup is learning from demonstration (LfD) [2, 3, 14, 18, 22, 29], where demonstrations are collected either by manually actuating the robot, i.e., kinesthetic demonstrations, or controlling it via teleoperation. LfD involves learning a policy from such demonstrations with the hope that it would generalize to new location/poses of the objects in unseen scenarios. Some recent works explore a relatively general version where a robot learns to imitate a video of the demonstration collected from either the robot's viewpoint [17] or with only a little different expert viewpoint [28].

In this paper, we tackle the generalized setting of learning from third-person demonstrations. Our agent first observes a video of a human demonstrating the task in front of it, and then it performs that task by itself. We do not assume any access to the state-space information of the environment and learn directly from raw camera images. To be successful, the robot needs to translate the observed goal states to its own context (imagine the goals in its viewpoint) as well as map the third-person actions to its trajectory. One way to solve this would be to use classical vision methods that estimate

---

Work done at CMU and UC Berkeley. Correspondence to pratyuss@csail.mit.edu

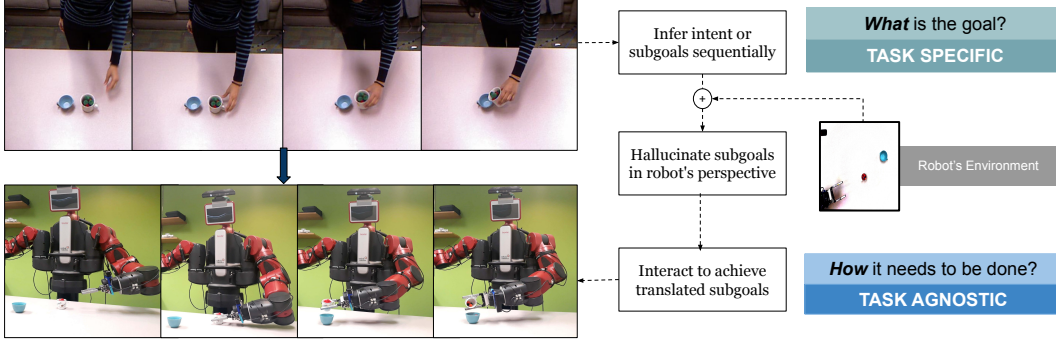

Figure 1: We study the general setup of learning from demonstration with of goal of building an agent that is capable of imitating a single video of human demonstration to perform the task with novel objects and tasks. The figure shows an example of a third-person video demonstration on top and the robotic agent trying to imitate the setup with objects in front. As shown on the right, our approach is to decouple the learning process into a hierarchy of *what* (high-level) module to translate the third-person video to first-person sub-goals and *how* module (low-level) to achieve those sub-goals.

location/pose of objects as well as the human expert and then map the keypoints to robot actions. However, hard-coding the correspondence from human keypoints to robot morphology is often non-trivial, and this overall multi-stage approach is difficult to generalize to unseen object/task categories. Another way is to leverage modern deep learning algorithms to learn an end-to-end function that goes from video frames of human demonstration to output the series of joint angles required to perform the task. This function can be trained in a supervised manner with ground truth kinesthetic demonstrations. However, unfortunately, today's deep learning vision algorithms require millions of images for training. While recent approaches [28] attempt to handle this challenge via meta-learning, the models for each of the tasks are separately trained and difficult to generalize to new tasks.

We propose an alternative approach by injecting hierarchical structure into the learning process in-between inferring the *high-level* intention of the demonstrator and learning the *low-level* controller to perform the desired task. We decouple the end-to-end pipeline into two modules. First, a high-level module that generates goal conditioned on the human demonstration video (third-person view) and the robot's current observation (first-person view). It predicts a visual sub-goal in the first-person view that roughly corresponds to an intermediate way-point in achieving the intended task described in the demonstration video. Generating a visual sub-goal is a difficult learning problem and, hence, we employ a conditional variant of Generative Adversarial Networks (GANs) [9] to generate realistic rendering [9, 11, 12, 16]. Second, a low-level controller module outputs a sequence of actions to achieve this visual sub-goal from its current observation. Both the modules are trained in a supervised manner using human videos and robot joint angles trajectories, which are paired (with respect to objects and tasks) but unaligned (with respect to time sequence). Our overall approach is summarized in Figure 2. The key advantage of this modular separation into task-specific goal-generator and task-independent low-level controller is that it improves the efficiency of our approach; how? The data-hungry low-level controller is shared across all tasks allowing it: (a) to be sample-efficient (in terms of data required per task) (b) robust and avoid overfitting.

We show experiments on a real robotic platform using Baxter across two scenarios: pouring and placing objects in a box. We first systematically evaluate the quality of both the high-level and low-level modules individually given perfect information on held-out test examples of human video and robot trajectories. We then ablate the generalization properties of these modules across the same task with different scenarios and different tasks with different scenarios. Finally, we deploy the complete system on the Baxter robot for performing tasks with novel objects and demonstrations.

## 2    Problem Setup: Third Person Visual Imitation

Consider a robotic agent observing its current observation state $s_t$ at time $t$. The action space of the robot is a vector of joint angles, referred to as $r_t$. Let $I_H$ be the sequence of images $h_t$ (i.e., video) of a human demonstrating the task as observed by the robot in third-person view, i.e., $I_H \equiv (h_0, h_1, ..h_T)$. Our goal is to train the agent such that, at inference, it can follow a video of a

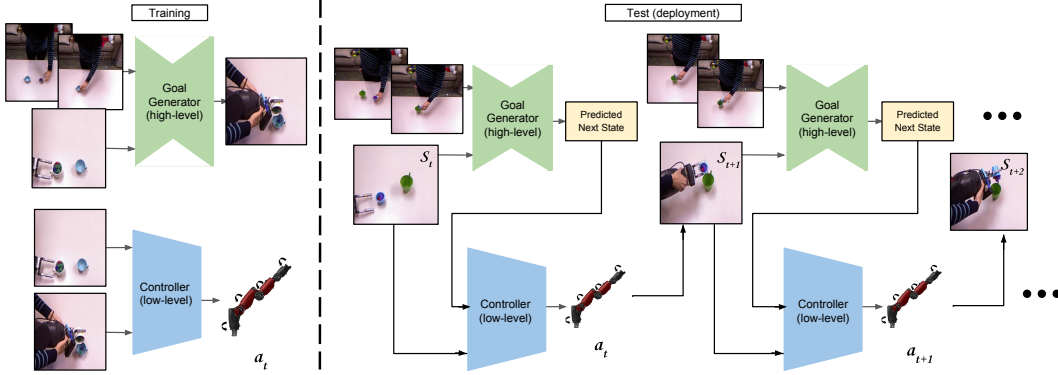

Figure 2: **Decoupled Hierarchical Control for Third-Person Visual Imitation Learning**: We introduce a hierarchical approach consisting of a goal generator that predicts a goal visual state which is then used by the low-level controller as guidance to achieve a task. [Left] During training, the decoupled models are trained independently. The goal generator takes as input the human video frames $h_t$ and $h_{t+k}$ along with the observed robot state $s_t$ to predict the visual goal state of the robot at $t + k$. The low level controller is trained using $s_t, a_t, s_{t+1}$ triplets. [Right] At inference, the models are executed one after the other in a loop. After reaching the current goal, the goal generator uses the new observed state $s_{t+1}$ and the next images of the human video to generate a new goal for the low-level controller to attain.

novel human demonstration video $I_H$ starting from its initial state $s_0$ by predicting a sequence of joint angle configurations $I_R \equiv (r_0, r_1, \ldots, r_T)$.

Our goal is to learn an agent that can imitate the action performed by the human expert in the third person video. We want to imitate only from raw pixels without access to full-state information about the environment. At training, we have access to a video of the human expert demonstration for a object manipulation task $I_H \equiv (h_0, h_1, ..h_T)$, a video of the same demonstration performed kinesthetically using the robot joint angle states $I_R \equiv (r_0, r_1, ..r_T)$ and a time series of the sequence of robot's first-person image observations $\tau_R \equiv (s_0, s_1, ..s_T)$. We leverage a recently released dataset of human demonstration videos and robot trajectories [25] where the demonstrations and trajectories are paired, but not exactly aligned in time. We sub-sample the robot and human demonstration sequences, which helps them roughly get aligned. In our setup, we have access to all the three time-series data at the training time, but only the time series data corresponding to the human demonstration image sequence at the test time. The other two time series would be predicted or generated by our algorithm.

## 3 Hierarchical Controllers for Imitation

An end-to-end model that goes from human demonstration video and robot's current observation to directly predict the robot trajectories would require a lot of human demonstrations. Instead, we inject the structure into the learning process by decoupling the imitation signal into **what** needs to be done from **how** it needs to be done. Decoupling makes our approach modular and more sample efficient than end-to-end learning. It also enables the system to be more interpretable, as the goal inference is now disentangled from the control task allowing us to visualize the intermediate sub-goals.

Our approach consists of a two-level hierarchical modular approach. The high-level module is a goal generator that infers the goal in the pixel space from a human video demonstration and translates it into what it means in the context of the robot's environment in the form of a pixel level representation. The second step is an inverse controller, which follows up on the generated cues from the visual goal inference model and generates an action for the robot to execute. These models are trained independently, and at test time, they are alternatively executed for the robot to accomplish the multi-step manipulation task, as illustrated in Figure 2.

### 3.1 High-Level Module: Goal Generator

The role of the high-level module is to translate the human demonstration images to generate sub-goals images in a way that is understandable to the robot. This high-level goal-generator could be learned

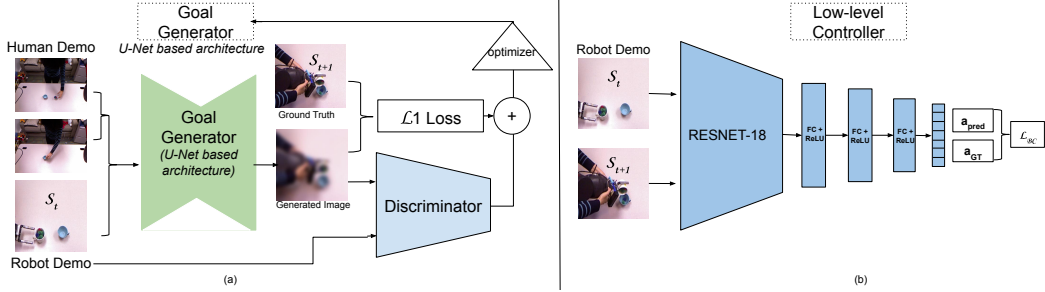

Figure 3: **(a) The Goal Generator**: The high-level goal generator network $\pi_H(.)$ takes as input the frames of the human demonstration video $h_t, h_{t+k}$ and the current observed state of the robot $s_t$ at time $t$. It is trained to generate the visual representation $s_{t+k}$ of the robot at time $t + k$. Instead of the complex goal image generation problem, our setup reduces the setup into a simpler *re-rendering* problem, i.e., move the pixels of robot image in the similar to the change in human demonstration images. **(b) Low-level Controller**: The inputs to the low-level controller are the observed state of the robot $s_t$ and goal state of the robot $s_{t+1}$. The model is trained to output the action $(a_t)$ that will cause it to transition to the goal state from $s_t$.

by leveraging the paired examples of human demonstration video and the robot demonstration video from our training data. The most straightforward formulation is to express the goal-generator as image translation, i.e., translating human demonstration image to robot demonstration. Image translation is a well-studied problem in computer vision and approaches like Pix2Pix [11], CycleGAN [31] could be directly deployed as-is. However, the stark difference between human and robot demonstration images is in terms of viewpoint (third-person vs. first person) and appearance (human arm vs. robotic arm) which makes these models much harder to train, and difficult to generalize as shown in Section 6.

We propose to handle this issue by translating *change in the human demonstration image* instead of the image itself. In particular, we task the goal-generator to translate the current robot observation image in the same manner as the corresponding human demonstration image is translated into the next image in sequence. This forces the goal-generator to focus on how the pixels should move (*re-rendering*) instead of figuring out the way harder task of generating the entire pixel distribution in the first place (*generation*). An illustration is shown in Figure 3. Further, in order to generate realistic looking sub-goals, we represent goal-generator via a conditioned version of generative adversarial networks with a U-Net [20] style architecture [9, 11, 12, 16].

At any particular instant $t$, the input to the goal generator model $\pi_H(.)$ is the visual state of the robot $s_t$ as well as the visual states of the human demonstration $h_t$ and $h_{t+n}$. This model is trained to generate the visual state of the robot at the $(t + n)^{th}$ step which can be represented as $s_{t+n}$. The overall optimization is as follows:

$$\min_{\pi_H} \max_{D} \quad \mathbb{E}_{s \in \mathcal{S}}[\log(D(s))] + \mathbb{E}[\log(1 - D(\pi_H(h_t, h_{t+n}, s_t)))] + \lambda \|\pi_H(h_t, h_{t+n}, s_t) - s_{t+n}\|_1$$

where $D$ refers to the GAN discriminator classification network, state $s$ is sampled form the set $\mathcal{S}$ of real robot observations from the training data, and the triplet $\{h_t, h_{t+n}, s_t\}$ are randomly sampled from the time series data of human demonstration and corresponding robot observations. In practice, we resort to using a wider context around the human demonstration images, for instance, more frames surrounding $h_t$ and $h_{t+n}$ especially when the human and robot demonstrations are not aligned. The L1-loss ensures that the correct frame is generated while the adversarial discriminator loss ensures the generated samples are realistic [16].

### 3.2 Low-Level Module: Inverse Controller

The main purpose of the low-level inverse controller is to achieve the goals set by the goal generator. The low-level inverse controller, $\pi_L(.)$, takes as input the present visual state of the robot demonstration $(s_t)$ along with the predicted visual state of the robot demonstration for the next time step $(\hat{s}_{t+n} = \pi_H(h_t, h_{t+n}, ))$ to predict the action that the robot should take to make the transition to its next state $(\hat{s}_{t+n})$. Since the task we test on may be performed by the left or the right hand of the robot depending on the human demonstration, we concatenate the seven joint angle states of the left as well as the right hand of Baxter robot. In our case, the predicted action is a 14-dimensional tuple of the joint angles of the robot's arms. The inverse model uses spatial information from the images

of the present visual state of the robot and the generated goal visual state to predict the action. The network used is inspired by the ResNet-18 model [10] and is initialized with the weights obtained from pretraining the network on ImageNet. An illustration of our controller is shown in Figure 3.

Note an exciting aspect of decoupling goals from the controller is that the controller need not be specific to a particular task. We can share the inverse controller across the different types of tasks like pouring, picking, sliding. Further, another advantage of decoupling goal inference from the inverse model is the ability to utilize additional self-supervised data ($r_t$, $r_{t+1}$, $s_{t+1}$ pairs) which does not have to rely on only perfectly curated demonstrations for training. We leave the self-supervised training for future work.

### 3.3 Inference: Third-person Imitation

At inference, we run our high-level goal-generator and low-level inverse model in an alternating manner. Given the robot's current observation $s_t$ and the human demonstration sequence $I_H$, the goal-generator $\pi_H(.)$ first generates a sub-goal $\hat{s}_{t+n}$. The low-level controller $\pi_L(.)$ then outputs the series of robot joint angles to reach the state $\hat{s}_{t+n}$. This process is continued until the final image of the human demonstration.

## 4 Implementation Details and Baselines

**Training Dataset**  We use the MIME dataset [25] of human demonstrations to train our decoupled hierarchical controllers. The dataset is collected using a Baxter robot and contains pairs of 8260 human-kinesthetic robot demonstrations spanned across 20 tasks. For the pouring task, we train on 230 demonstrations, validate on 29, and test on 30 demonstrations. For the models trained on multiple tasks, 6632 demonstrations were used for training, 829 for validation, and 829 for test. In particular, each example contains triplet of human demonstration image sequence, robot demonstration images, and robot's joint angle state, i.e., $\{(h_0, h_1, ..h_T), (\hat{r}_0, \hat{r}_1, \ldots, \hat{r}_T), (s_0, s_1, ...s_T)\}$. We sub-sampled the trajectories (both images and joint angle states) to a fixed length of 200 time steps for training our models. For training low-level inverse model, we perform regression the action space of robot $a_t$ which is a fourteen dimensional joint angle state $[\theta_1, \theta_2, \theta_3..., \theta_{14}]$. All the training and implementation details related to our hierarchical controllers are provided in the supplementary.

**Baseline Comparisons**  We first perform ablations of our modules and compare them to different possible architectures, including CycleGAN [31], and L1, L2 loss based prediction models. We then compare our joint approach to two different baselines: (a) End-to-end Baseline [25]: In this approach, both the task of inference and control are handled by a single network. The inputs to the network are consecutive frames of the human demonstration around a time step t, along with the image of the robot demonstration at the time step t. The network predicts the action that the robot must then take at time step t to transition to its state at time step t+1. (b) DAML [28]: The second baseline, we compare our results with is the Domain Adaptive Meta-Learning (DAML [28]) baseline. The algorithm is targeted for recovering the best network parameters for a task via a single gradient update at test time using meta-learning.

## 5 Results: Generalization of Individual Hierarchical Modules

The hierarchy modules run alternatively at test time, and hence, each model relies on the other's performance at the previous step. Therefore, in this section, we evaluate the generalization abilities of both of our individual modules of the hierarchy while assuming ground truth access to others. We evaluate top-level goal generators assuming the inverse model is perfect and evaluate the inverse-model assuming access to perfect goal-generator. We study generalization across three different scenarios: new location, new objects, and new tasks.

### 5.1 Generalization to new positions of the same object

**Goal Generator:** The ability to condition inferred goals in the robot's own setting is a crucial aspect of our approach. The sensitivity analysis of the goal generator with respect to the position of the objects can help us understand how well the goal generator generalizes in terms of object positions.

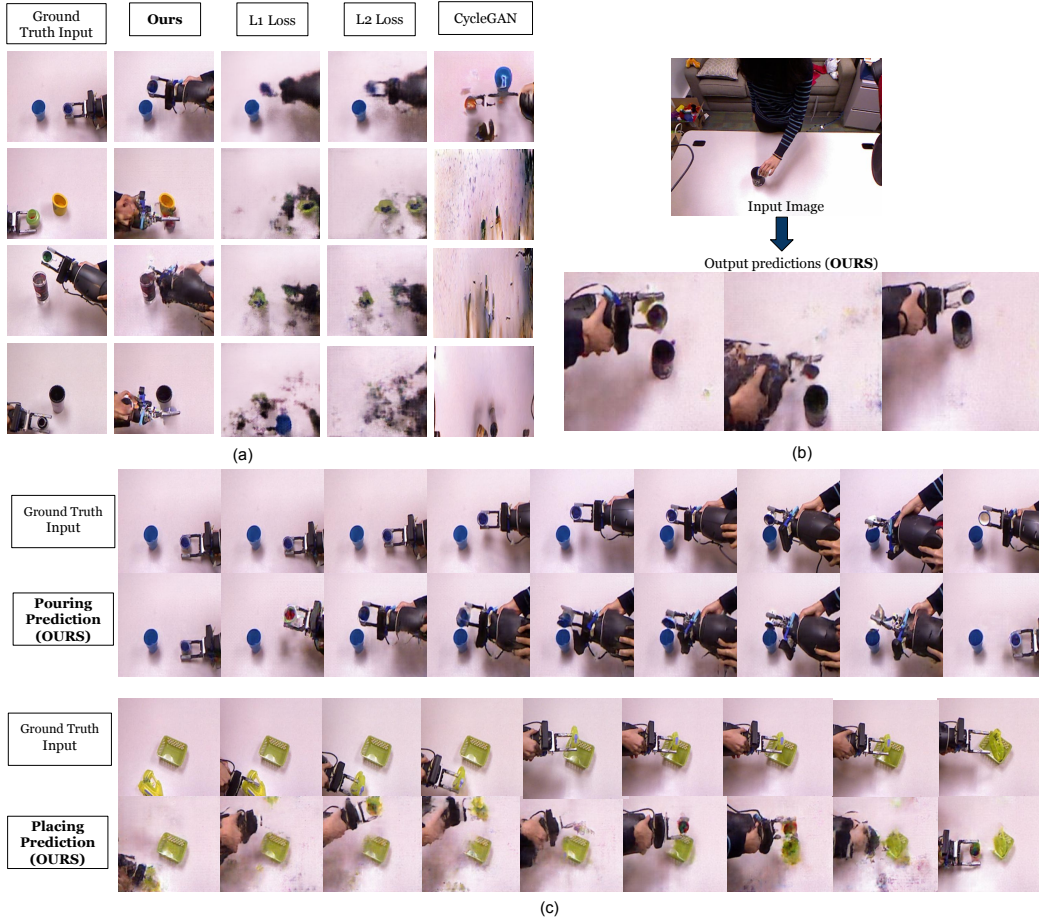

Figure 4: **(a) Goal Generator Comparison**: The predictions of the outputs generated by the goal generator when optimized using different methods. Our model, which is trained to translate the robot's current image instead of generating from scratch, generates the sharpest and accurate results. **(b) Sensitivity Analysis of the Goal Generator**: Given the input human demonstration of a task, we test the sensitivity of goal-generate wrt object locations. Our model can hallucinate accurate sub-goals in accordance with the object location. **(c) Goal Generator Predictions**: The images in the first row are the input observed robot states. The second row contains goals generated by the goal generator from the input images. The predictions are at an interval of ten steps (approx. 2sec) ahead into the future. As shown, predicted sub-goals are consistent across the trajectory.

In Figure 4 (b), we show a scenario where the input of the human demonstration is fixed, but the positions of the objects are varied at test time. The predictions of the goal generator reveal that it is responsive in accordance with change in object positioning. A quantitative analysis of this positional generalization is performed jointly with the evaluation of generalization ability to new objects in Table 1.

**Inverse model:** To check the ability of the inverse model to generalize to new positions (given perfect goal-generator) of the object, we test the inverse model using ground truth images of the test set. This quantitative evaluation is performed jointly with the evaluation of generalization to novel object in Table 2 and discussed in the next sub-section.

## 5.2    Generalization to new objects

We now evaluate the ability of our models to generalize manipulation skills to unseen objects.

**Goal Generator:** Figure 4(a) shows the ability of the goal generator to generate meaningful sub-goals given a demonstration with novel objects. A quantitative evaluation is shown in Table 1 for the goal generation ability when tested with novel objects in different configurations. Our approach

| Method | L2 | L1 | PSNR | SSIM |
|---|---|---|---|---|
| L1 only | 60.24 | 72.57 | 2.92 | 0.15 |
| L2 only | 76.44 | 75.94 | 3.02 | 0.14 |
| Cycle GAN [31] | 99.15 | 118.67 | 2.37 | 0.11 |
| **Goal-Gen(Ours)** | **39.98** | **52.37** | **3.95** | **0.18** |

Table 1: Goal-Generator generalization to novel objects and locations. Our goal generator outperforms the other approaches, both qualitatively and quantitatively, across different loss metrics. The models are evaluated on the pouring test set.

| Method | RMSE (mean) | RMSE (stderr) |
|---|---|---|
| End to End (all) [25] | 14.7 | 2.3 |
| End to End (single) [25] | 8.9 | 1.7 |
| DAML (single) [28] | 11.84 | 2.1 |
| **Ours (all)** | **14.4** | **2.2** |
| **Ours (single)** | **8.1** | **1.6** |

Table 2: Inverse model generalization to novel objects and locations. This table contains models trained on all tasks of the MIME dataset (all) and just the task of pouring (single). The models are evaluated on the common test set of pouring

outperforms the baselines on all four metrics and generalizes better to new objects both quantitatively (Table 1) and qualitatively (Figure 4(a)). In addition to the baselines shown in Table 1, we also tried an optical flow baseline which did not perform well and was unable to account for in-plane rotations that the task like pouring required. The performance is (L1: 127.28, SSIM:0.81) significantly worse than other methods.

**Inverse model:** A quantitative evaluation of generalization to new objects and locations is shown in Table 2. Our model outperforms all other baselines by a significant margin. The generalization to diverse positions of objects of the inverse model can be attributed to its training across many different positions of diverse objects.

In addition to the baselines in Table 2, we also compare against the two feature matching based approaches. First, we compute trajectory-based features of the frames of human demonstration and then find the nearest neighbors from the other demonstrations in the training set. The joint angles corresponding to the nearest demonstrations are then considered as the prediction. The trajectory-based features were computed using state-of-the-art temporal deep visual features trained on video action datasets [4]. Using these features as keys to match the nearest neighbors resulted in a rMSE of 22.20 with a stderr of 2.14. Secondly, we used a static feature-based model where we align human demonstration frames with robot ones in SIFT feature space. This resulted in a rMSE value of 45.32 with a stderr of 6.12. Both the baselines perform significantly worse than our results shown in Table 2. In particular, SIFT features did not perform well in finding correspondences between the human and robot demonstrations because of the large domain gap.

## 5.3 Generalization to new tasks

So far, we have tested generalization with respect to objects and their positions. We now evaluate the ability of our approach to generalize across tasks.

**Goal Generator:** The goal generator is not task-agnostic. We leave training a task-agnostic goal generator for future work. In principle, since both the goal generator and inverse model don't depend on temporal information, it should potentially be possible to train a task-agnostic Goal Generator.

**Inverse Model:** The inverse model is not trained to perform a particular task. No temporal knowledge of trajectories is used while training the module. This ensures that while the model predicts every step of the trajectory it doesn't have any preconceived notion about what the entire trajectory will be. Hence, the role of low-level controller (inverse model) is decoupled from the intent of the task (goal-generator) making it agnostic to the task. The ability of the model to generalize to new tasks is demonstrated in Table 3. We train on the first 15 tasks from MIME dataset and test on a held-out dataset for

| Method | Train (15 Tasks) | | Test (5 Tasks) | |
|---|---|---|---|---|
| | Mean | Stderr | Mean | Stderr |
| End to End [25] | 23.63 | 1.06 | 24.83 | 1.56 |
| DAML [28] | 35.90 | 1.56 | 36.45 | 1.55 |
| **Inv. Model (Ours)** | **18.05** | **0.76** | **16.90** | **1.04** |

Table 3: Generalization of the Inverse-Model to New Tasks. Our inverse model is trained on 15 tasks of the MIME dataset. It is evaluated on a held-out set from training tasks as well as 5 novel tasks where it significantly outperforms the baselines.

15 training as well 5 novel tasks. Our model has a much lower error on both the trained tasks as well as the novel tasks than the baseline methods. We want to note that DAML [28] is a generic approach, not mainly designed for task transfer in third person, and the results in the original paper have been shown in the context of single planar-manipulation tasks. It has not been shown to scale to training

on multiple task categories together. Hence, further changes might be required to scale DAML for transfer across tasks.

# 6   Results: Generalization and Evaluation of Joint Hierarchical Model

The final test of our approach is to evaluate how the decoupled models perform when run together. Robot demo videos are on the project website https://pathak22.github.io/hierarchical-imitation/.

We look at two tasks - Pouring and Placing in a box. In the task of pouring, the robot is required to start at a given location and then move to a goal location of the cup that needs to be poured into. This task requires the model to predict the different parts of the task correctly which are reaching the goal cup and pouring into it. Since the controller of the robot is imperfect and the predictions can be slightly noisy, we consider a reach to be successful if the robot reaches within 5cm of the cup. Similarly, we consider pouring to be successful if the robot reaches and does the pouring action in 5cm radius of the cup. These evaluation metrics are similar to those used by Yu et al. [28].

For the task of placing in the box, we categorize a successful placing in a box if the robot is able to reach within 5cm of the box and is then able to drop the object within 5cm of the box. Further, the models are trained on the task on pouring alone and we evaluate how they generalize to the task of placing.

| Method | Pouring | | Placing | |
|---|---|---|---|---|
| | Reaches | Pours | Reaches | Drops |
| End to End [25] | 20% | 8% | 20% | 10% |
| DAML [28] | 25% | 15% | 20% | 10% |
| **Hierarchy (Ours)** | **75%** | **60%** | **70%** | **50%** |

Table 4: Joint evaluation of our hierarchical decoupled controllers. Our approach outperforms the other baselines on the tasks of pouring and placing in a box with a significant margin, however, it is still much far from perfect completion of the task.

For the high-level goal generator, it is crucial to generate good quality results over a long horizon to ensure the successful execution of the task. Our approach of using a goal generator to predict high-level goals and an Inverse model to follow up on the generated goals in alternation outperforms the other approaches, as shown in Table 4. The test sets comprised of demonstrations with novel objects placed in random locations. The test not only required the individual models to generalize well but also works well in tandem with the possibility of imperfect predictions and actions from one another.

# 7   Related Work

Inferring the intent of interaction from a human demonstration and successfully enabling a robot to replicate the task in it's own environment ties to several related areas discussed as follows.

**Domain Adaptation:** Addressing the domain shift between the human demonstrator and robot (e.g., appearance, view-points) is one of the goals of our setup. There has been previous work on transfer in visual space [11, 30] and on tackling domain shift from simulation environments to the real-world [5, 15]. Some of these approaches map data points from one domain to another [11, 30]. Other approaches aid the transfer by finding domain invariant representations [21, 26]. Along similar lines, Sermanet et al. [24] looks at learning view-point invariant representations that are then used for third-person imitation. Training such a system would require training data with videos collected from multiple viewpoints. Moreover, learning task-invariant features might not alone be enough to aid the transfer to the robot's setting because of the differences in the physical configurations. Our approach handles these issues via modular controllers.

**Learning from Demonstrations (LfD):** LfD generally uses demonstrations obtained from trajectories collected by kinesthetic teaching, teleoperation, or using motion capture technology on the robot arm [2, 3, 14, 18, 22, 29]. LfD has been successful in learning complex tasks from expert human trajectories, for instance, playing table-tennis [13], autonomous helicopter aerobatics, and drone flying [1]. Most of these focus on learning a task from a handful of expert demonstrations for a single task. Our goal is to start by using demonstration data collected across some objects and tasks but enable the robot to imitate the task by just watching one video of a human demonstrating the task with new objects.

**Explicitly Inferring Rewards:** Other approaches explicitly infer the reward associated with performing a task from the human demonstrations through techniques such as inverse reinforcement learning [19, 23]. The rewards become representations of the sequence of goals of the task. After construction of the reward functions, the robot is trained using reinforcement learning by collecting samples in its environment to maximize the reward. However, such systems end up needing significantly large amounts of real-world data and have to be re-trained for every new task from scratch, which makes them difficult to scale in the real world. In contrast, our supervised learning approach is trained via maximum likelihood, and thus, efficient enough to scale to real robots.

**Visual Foresight:** Visual foresight has been popular for self-supervised robot manipulation [6–8, 27], but it relies on task specification in the form of dots in the image space and are action conditioned visual space predictions. Our setting relies on no hand specified goals. The goals in our setting are specified from the human demonstration videos directly. This flexibility lets us specify harder tasks such as pouring, which would have been difficult to specify from dots on images alone.

## 8  Discussions

We present decoupled hierarchical controllers for third-person imitation learning. Our approach is capable of inferring the task from a single third-person human demonstration and executing it on a real robot from first-person perspective. Our approach works from raw pixel input and does not make any assumption about the problem setup. Our results demonstrate the advantage of using a decoupled model over an end-to-end approach and other baselines in terms of improved generalization to novel objects in unseen configurations.

Future Directions: Our high-level and low-level modules currently operate at a per-time step level and don't make use of temporal information, which results in the predicted trajectories being shaky. A naive inverse controller modeled via LSTM could incorporate the temporal information but it easily learns to cheat by memorizing the mean trajectory making it hard to generalize to novel tasks. However, training on lots of tasks together could potentially alleviate this limitation. An added advantage of the explicit decoupling of the models is the ability to utilize additional self-supervised data to train the low-level controller and make it robust to failure and different types of joint configurations. We leave these directions for future work to explore.

## Acknowledgements

We would like to thank David Held, Aayush Bansal, members of the CMU visual learning lab and Berkeley AI Research lab for fruitful discussions. The work was carried out when PS was at CMU and DP was at UC Berkeley. This work was supported by ONR MURI N000141612007 and ONR Young Investigator Award to AG. DP is supported by the Facebook graduate fellowship.

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
