[Supplementary Material]

# Supplementary Material:
# Third-Person Visual Imitation Learning via Decoupled Hierarchical Controller

**Pratyusha Sharma**
MIT

**Deepak Pathak**
Facebook AI Research

**Abhinav Gupta**
CMU

## 1 Implementation Details

### 1.1 Goal Generator (high-level)

The goal generator uses pix2pix[2] inspired framework. The generator network is a U-Net 128 block with skip connections between the $i^{th}$ and $(m-1)^{th}$ layers where $m$ is the number of layers in the U-Net block. The encoder and decoder architecture are as shown in Figure 3 of the main paper. The input to the model is an image of shape $(128X128)$. The images are randomly jittered by resizing to 140X140 and then cropped back to 128X128. The network is optimized using Adam [3] with a learning rate of 0.0002 along with momentum parameters $\beta_1 = 0.5, \beta_2 = 0.999$.

The input to the network contains the human demonstration image at time step $t$ and $t + k$ ($h_t$ and $h_{t+k}$) along with the robot demonstration image at time step $t$ ($s_t$). The output of the network is the robot goal state ($\hat{s}_{t+k}$) at time $t + k$. While we want precise goal predictions which would require the long multi-step task to be broken into smaller steps, we also require the goal generator to predict goals that look significantly different from the current observed state $s_t$ so the inverse controller can predict a change in state. Empirically, we find that after subsampling the trajectories to 200 time steps a value of $k = 10$ handles this trade-off best.

### 1.2 Inverse Controllers (low-level)

The inverse model or the local controller consists of 4 convolution blocks of ResNet-18 [1] followed by three fully connected layers. The ResNet blocks are initialized with pre-trained weights on ImageNet. The input to the network was the robot state at time $t$ and the goal state $t + 1$. The action predicted by the network was a fourteen-dimensional tuple of the joint angle states of the different joints of both the left and right arms of Baxter, $[\theta_1, \theta_2, \theta_3..., \theta_{14}]$. The input images were jittered by random cropping 85% of the image to make the model robust to vibrations in the robot arms and camera. The learning rate used to train the model was 0.001 and the optimized using Adam [3].

## 2 Generalization of Inverse Model: Simulation Experiments

In addition to our real-world experiments discussed in Section 5.2 of the main paper, we also trained an inverse model in simulation with the Sawyer robot. The trajectories used to train Sawyer were obtained from a policy trained on reaching with different objects placed in front of it. Demonstrations were created by training policies using proximal policy optimization(PPO). The policies were trained on a diverse set of objects to collect 500 demonstrations. For different object locations on new objects at test time, our learned controller achieves mean RMSE of 6.09 with a stderr of 2.8, which suggests the robustness of the controller.

---

Work done at CMU and UC Berkeley. Correspondence to pratyuss@csail.mit.edu