[Reviews · NeurIPS 2019]

Reviewer 1



This paper proposes a technique for a robot to learn to imitate a novel task from a single video demonstration. The model consists of two levels of hierarchy: intent modeling from a third-person video and a low-level controller. The paper is well-written and easy to follow. However, rather than restricting the discussion to specific techniques, it would have been interesting if the paper broadly discusses the concepts. For instance, when describing the method it is important to mention the problem/requirement and then provide the rationale for choosing a particular method. The main theme of the paper is decoupling the hierarchy and learning. In fact, the idea of running robots in a modular fashion is the defacto choice in almost all robotics tasks, though there is a negligible minority of roboticists attempting to learn robotic tasks in an end-to-end fashion. The low-level controllers in robotics are always task-agnostic. Therefore, I do not see any novelty in the concept of decoupling “what” and “how.” It is also worth noting that there is a rich literature on hierarchical reinforcement learning. What is the advantage of directly predicting the images over learning an abstraction of the intent and then trying to predict the position/configuration of the end-effector? Why do we want to explicitly represent the next state prediction as an image? Why cannot it be a latent feature vector, eschewing the requirement of a conventional GAN that is designed to produce photo-realistic images? Drawing similarities to how an intelligent animal would imitate, I do not think they explicitly hallucinate as images. This might open the door to other generative modeling techniques such as VAEs, those based on optimal transport, and normalizing flows. I would like to know the authors' thoughts on using DiscoGAN, I2I, MUNIT, and similar generators. Further, how can we capture the uncertainty in inferring the intent? It is not quite true that the robot learns novel tasks from a single video demonstration because the low-level controller, as well as the goal generator, have been pre-trained on various tasks. Can the low-level controller perform tasks that it has not been trained on? With reference to section 5.3, can we have a principled quantitative approach to define tasks and perform sensitivity analysis? Although picking and sliding are somewhat different tasks for a human, they might be/not be similar for a mathematical model. Since this is a modeling paper, the emphasis should not be running experiments on a real robot, but performing rigorous experiments, even on a simulator, to validate the model and analyze the sensitivity to various human demonstrations. I would personally appreciate a scientifically valid “design and analysis of experiments” (DOE) over demonstrations. It is always a good idea to avoid adding website links such as google sites as they can be used to track the location/IP address of the reviewers. The provided website provides no purpose as it only contains a video. Please consider attaching the video instead. POST-REBUTTAL COMMENTS: Thank you for the interesting rebuttal! After going through the rebuttal and other reviews, I have three main concerns about the paper. 1. Lack of rationale The paper discusses what was done, not why it was done---the most important question in science. "which did not perform well" - I wasn't expecting a numerical comparison with all possible generative models (thanks for the comparison, though!). Knowing the motivation and rationale for using GANs is important. In the rebuttal, rather than simply saying that it did not work, it is important to tell us why it did not work (at least a hypothesis). 2. The way the novelty is presented - a minor concern The novelty of the paper is applying a set of standard ML techniques for third-person visual imitation learning, not decoupling the hierarchy. The latter is highlighted as the novelty throughout sections 1, 2, and 3. 3. Experiments - a major concern Real-robot experiments are extremely important. The message I attempted to convey was the importance of "controlled experiments" which help to understand the capacity of the proposed algorithms. For instance, consider the following experiment. *Experiment*: We need to understand the "space of demonstrations" the proposed algorithm is valid because human demonstrations can be varied. We asked 100 people (is 100 enough? - "experimental power" in DoE) to perform pouring in this particular experimental setup. We isolated X and Y factors that could confound. (Alternatively, we used a standard dataset that contains pouring actions--I don't think there is one). Then, we found out that the proposed algorithm is valid for pouring actions with these particular class of demonstrations. Merely by running this experiment, we know when this algorithm works and know what improvements we need to make in the next iteration of the work. What this paper suggests is a concept and therefore what it should attempt to do is proving that the concept is valid and have a great potential. What are given in the paper and rebuttal are demonstrations, not experiments. No details about the experimental setup and conditions are provided. Numerical results are less useful without having a standard framework or controlled environments. “manipulation tasks involve intricacies like fine-grained touch, slipping, real objects” - This is exactly why I emphasized the importance of simulations so that we can isolate these artifacts and try to understand our algorithm: why it works and when it works. Otherwise, we might have to run thousands of experimental evaluations to show that our algorithm is indeed valid in the presence of these artifacts.

Reviewer 2



The paper is well-written, clear and approachable. It would be much stronger if it was backed by open-source code since it’s far from solving the problem, and could be improved on in several ways, including ones suggested in the video (temporal modeling). I would also have been interested in the claims that the two models can be trained completely independently, since, in the absence of randomization of the camera and robot setup, it seems unlikely that the system as-is would have learned much robustness to calibration. I understand that stating this may break anonymization, but am I correct to suspect that the robot used was the same one as in [22]? If so, it would be good to address directly the question of whether the two trained models have to be trained on very tightly coupled data or not. POST-REBUTTAL: Thanks to the authors for their responses. I took note that the code and data would be open-sourced, which strongly increase the value of the publication in my POV. I also note that the experiments transferring across robots mitigate my concern about overfitting to that one particular setup. Increasing my confidence level.

Reviewer 3



The paper presents a framework for learning by demonstration using third person videos. The method is based on decoupling the intended task from the controller, by learning a hierarchical setup where the high-level module generates goals conditioned on the third-person video demonstration for the low-level controller. Due to its modularity, the proposed approach is more sample efficient than other end-to-end approaches, and the learned low-level controller is more general. The paper is well written and well structured, it includes insightful figures and diagrams, and fair ablations and comparisons. Originality: the paper presents an interesting approach to use third-person views as demonstrations for a robot; learning from demonstrations, including from videos, is not a novel contribution, as well as learning modular controllers in the form of an inverse model. Despite limited novelty, the approach presented in this paper is neat and clear. It would be interesting to see comparisons for example with methods that explicitly find correspondences between the demonstrator and the robot, and with methods based on trajectory-based demonstrations. Quality: the submission is technically sound, and the approach explained in a clear way; the results (including those shown in the video) suggest that there is still room for improvement in terms of succeeding in completing the different tasks (e.g. the pouring policy execution seems wobbly and lucking robustness). Some discussion about the limitations of the proposed method would help evaluating the overall results. Clarity: the paper is overall clearly written and well organized; a discussion around the limitations of the proposed approach could be added. Significance: the paper provides an interesting way to address learning by third-person view demonstrations in robotics; this is a challenging and important field and this contribution is interesting with respect to more classical approaches based on hand-crafted models or task-specific solutions.

[Author Response · NeurIPS 2019]

We thank the reviewers for their helpful feedback. The reviewers found our "demonstrations on a real robot inter-esting"(R1); approach "novel and sound" and "backed by solid empirical results"(R2); and "bridging fields"(R3). The reviewers R1 and R3 suggested additional experiments. We are pleased to report that we have completed those experiments. We report those results and address other concerns below.

**[R1] "running robots in a modular fashion is the defacto choice... negligible minority of roboticists attempting to learn robotic tasks in an end-to-end... low-level controllers in robotics are always task-agnostic... do not see any novelty in the concept of decoupling what and how":** We tackle the generalized setting of learning from *third-person demonstrations from raw sensory data*. At test-time, our agent first observes a video of a human demonstrating the task in front of it, and then it performs the task by itself. To the best of our knowledge, such general setup of manipulating novel objects from raw-sensory data with 3rd person demos (and not kinesthetic trajectories) is not yet defacto for imitation learning.

Although, traditional robotics approaches employ modularity in terms of planning and control, but those controllers are based x,y coordinates or joint angle positions as input in a fully observable environment. Hence, they are difficult to generalize to unseen objects/orientations (as also noted by R3). In contrast, our low-level controller is more like a policy and *learn* from raw high-dimensional images which allows generalization to novel objects/configurations.

The only prior work that tackles third-person imitation from raw sensory observation without any handcrafted features is DAML [24] and we already compared our proposed approach to it. Since, the reviewer has not provided references, it is extremely difficult for us to argue or provide additional comparisons other than already in the paper.

**[R1] "...literature on hierarchical reinforcement learning":** Current RL approaches from pixels (not state vectors) usually take millions of steps in simulation [Minh et.al. 2016] and are too sample inefficient to be scalable to complex robotics scenarios. In contrast, our supervised learning approach is trained via maximum likelihood, and thus, efficient enough to scale to real robots.

**[R1] "Can the low-level controller perform tasks that it has not been trained on?":** Yes, indeed. Section 5.3, Table 3 shows that our learned controllers generalize to unseen tasks at test time significantly better than the baselines.

**[R1] "emphasis should not be running experiments on a real robot, but performing rigorous experiments, even on a simulator, to validate the model and analyze the sensitivity":** We respectfully disagree on the opinion for the emphasis not being real robots. Firstly, manipulation tasks involve intricacies like fine-grained touch, slipping, real objects etc which are still very difficult to simulate and major challenge for imitation learning. Secondly, real robots need the algorithm to be extremely sample efficient to be applicable.

Upon R1's suggestion, we setup a simulation environment where we transfer results from a Baxter robot demonstrations to a Sawyer robot. We inverse model trained on this simulation data. With respect to different object locations, our learned controller achieves mean RMSE of 6.09 with stderr of 2.8, which suggests the robustness of the controller.

**[R1] "... predicting images over latent features":** Our approach is agnostic to features/images, and our contribution is orthogonal to the design of observation space. Although, VAE-like methods can learn good appearance based embeddings, but learning an embedding that respects fine-grained displacements while capturing the task-oriented cues is an open research problem. One way could be to learn such an embedding via inverse model, but then it would become specific to the training task and prevent the modular decoupling. Hence, we opted for prediction in the image space.

**[R1] "... thoughts on using DiscoGAN... and similar generators":** DiscoGAN is very similar to CycleGAN baseline shown in Table 1. Upon suggestion, we also tried on flow-based models [Zhou et.al. 2016] which did not perform well and were unable to account for in plane rotations that tasks like pouring require. The performance is (L1: 127.28, SSIM: 0.81) significantly worse than our method in Table 1.

**[R2] "... open-source code":** We will release our code and data publicly with the paper.

**[R2] "whether trained models have to be trained on very tightly coupled data or not; robot same as [22]?":** We used a Baxter Robot for our experiments (as also employed in [22]). However, we re-calibrated the robot from scratch and tested with different objects. While similarity in distribution is always helpful, the modular decoupling is also reasonably effective when the data is not tightly coupled.

**[R3] "Add comparisons with feature-based models... with methods based on trajectory":** Upon reviewer's suggestion, we ran two baseline comparisons. (1) Trajectory-based features: Given a human demo at test time, we find feature-based nearest neighbor human demonstrations from training set and replay their corresponding joint angles. The feature used for matching are state-of-the-art temporal deep visual features trained on video action datasets (Non-local Neural Networks - [Wang et.al. 2018]); performance = [rMSE: 22.20, stderr: 2.14]. (2) Static feature based model: Align human demonstration frames with robot ones in SIFT space and find nearest neighbors from training; performance = [rMSE: 45.32, stderr: 6.12]. Both the baselines perform significantly worse than our results shown in Table 2. In particular, SIFT features didn't perform well in finding correspondences between the human and robot demonstrations because of the large domain gap. We will try ablations with more feature descriptors for the final version.

**[R3] "Add discussion about limitations of the approach":** This is a great suggestion and we will include a section on this. Some limitations include (a) Temporal continuity is not used in the trajectory prediction; (b) Our inverse model can be trained via self-supervised data, but goal-generator needs demonstration. (c) Goal-generator is not task-agnostic.

[Meta-Review · NeurIPS 2019]

This paper presents a method for imitation learning that decouples learning an image-based goal from the controller. This decomposition in itself is not novel, but the authors incorporate the latest neural network methods to generate the goal images and learn the controller. There was discussion among the reviewers about how to properly validate and compare the proposed method; everyone appreciated the fact that the work used a physical instantiation but there was concern about the lack of baseline comparisons. However, in the end, the PC concluded that the manuscript along with the latest revisions should be presented at NeurIPS.